# ShuffleNorm: A Better Normalisation for Semi-supervised Learning

## Abstract

We identify critical challenges with normalisation layers commonly used in fully supervised learning when applied to semi-supervised settings. Specifically, batch normalisation (BN) can experience severe performance degradation when labelled and unlabelled data have mismatched label distributions, due to biased statistical estimation. This results in unstable gradients, hindering the model's ability to converge effectively. While group/layer normalisation (GN/LN) avoids these issues, it lacks the stochastic regularisation provided by BN, leading to weaker generalisation. Poor generalisation, in turn, produces low-quality pseudo-labels, exacerbating confirmation bias. To address these limitations, we propose novel normalisation techniques termed Shuffle Layer normalisation and Shuffle Group normalisation (SLN/SGN) that introduce controllable randomness into LN/GN without increasing model parameters, thus making semi-supervised learning more robust and effective. Through experiments across diverse datasets, including image, text, and audio modalities, we demonstrate that SLN/SGN significantly enhances the performance of state-of-the-art semi-supervised learning algorithms.

## 1 Introduction

Semi-supervised learning aims to design a training scheme that enables deep learning models to achieve superior performance with minimal reliance on large amounts of labelled data. Typically, research on training schemes is model-agnostic, meaning we often use mainstream models from fully supervised learning to investigate the training scheme. However, are the modules of these models, originally proposed for fully supervised learning, truly suitable for semi-supervised learning? In this paper, we start by exploring the normalisation layers within models to answer this question.

Normalisation layers, which stabilise model training and accelerate convergence, are widely used in deep neural networks. For instance, in convolutional neural networks (CNNs), batch normalisation (BN) (Ioffe & Szegedy, 2015) is the most popular choice. However, these default normalisation layers are not necessarily optimal for semi-supervised learning, as our findings suggest.

In semi-supervised learning, a consistent label distribution for labelled and unlabelled subsets cannot be guaranteed, especially in real-world applications. We discovered that BN is highly susceptible to performance degradation when the label distribution of unlabelled data significantly deviates from that of labelled data. As shown in Fig. 1, the accuracy of an image classification model drops significantly as the label inconsistency ratio $r$ increases. One reason for this decline is that as $r$ increases, the amount of in-distribution data in the unlabelled subset decreases, while the proportion of noisy data increases. This inevitably leads to a performance drop. However, we found that this is not the only reason; another important factor is the increased upper bound of the gradient's difference due to biased statistical estimates in BN, *i.e.*, unsteady gradients. Therefore, the stable and efficient convergence of the model is no longer guaranteed. Group Normalisation (GN) (Wu & He, 2018) was proposed to solve the problem of small minibatch issues in BN initially. We found that GN and its special case, *i.e.*, Layer Normalisation (LN), inherently avoid this issue as the statistics used for normalising a data sample are independent of other samples. However, they have yet to surpass BN in many cases, especially when the batch size is sufficiently large (Wu & He, 2018). By analysing the operations of BN and GN/LN, we believe the performance gap stems from the missing stochastic regularisation in GN/LN. The absence of stochastic regularisation leads to

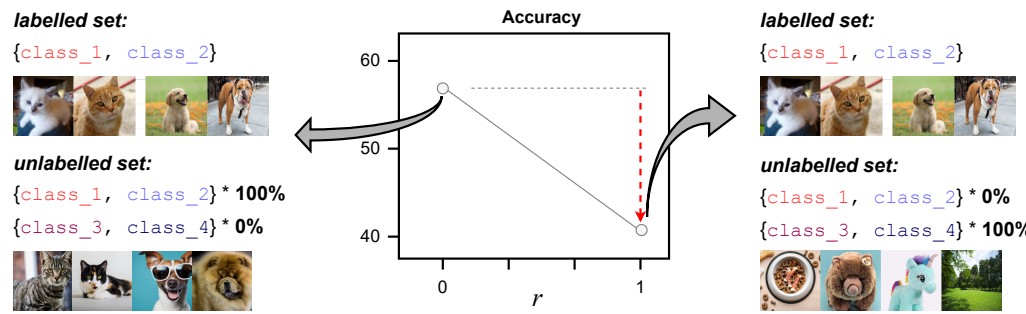

Figure 1: Performance drops as the label inconsistency ratio increases. $r$ is the label inconsistency ratio. If $r = 0$, there is no out-of-distribution data in the unlabelled subset. If $r = 1$, all data in the unlabelled subset are out-of-distribution.

inadequate model generalisation, causing the model to produce less accurate pseudo-labels in semi-supervised learning. Training with incorrect pseudo-labels inevitably exacerbates the confirmation bias problem (Arazo et al., 2020).

Therefore, we introduce controlled randomness into these normalisation layers, allowing them to retain their strengths while incorporating controllable stochastic regularisation. Our modification makes GN/LN a more effective choice for semi-supervised learning in CNNs and Transformers. Notably, our modifications do not add any additional learnable parameters, meaning pre-trained model parameters can still be used for initialisation. Moreover, the extra computational overhead introduced by our method is negligible. The proposed cost-free normalisation layers termed Shuffle GN (SGN) and Shuffle LN (SLN) solved the aforementioned performance degradation issue in the inconsistent label distribution scenario. Most importantly, they significantly improve the performance of state-of-the-art models with GN/LN in semi-supervised tasks across three modalities including image, text, and audio. For example, on the STL10 dataset, our normalisation layers increase the baseline performance by 3.7%.

In summary, our contributions are as follows:

- We identify the performance drop risk of BN in semi-supervised learning.

- By comparing the operation of BN and GN/LN, we propose a simple yet effective modification to GN/LN that introduces more randomness, which significantly improves the performance of the baseline models with GN/LN in semi-supervised learning.

- Our proposed SGN and SLN are fully compatible with existing pre-trained weights, allowing them to be applied to downstream tasks without retraining the backbone, with minimal computational overhead.

- We demonstrate the effectiveness of our method in semi-supervised learning tasks on image, text, and audio modalities.

## 2  RELATED WORKS

**Semi-supervised Learning** is targeting to optimise a model using a combination of low numbers of labelled and large amounts of unlabelled data. It alleviates the data-hungry problem in supervised training of deep learning models, and most importantly, it significantly contributes to the data engine of large-scale AI models such as SAM (Kirillov et al., 2023). Effectively learning recognition patterns with limited labels and leveraging the unlabeled data is essential to solving this problem. The main categories of algorithms in semi-supervised learning include: a) generative models, b) graph-based methods, and c) pseudo-labelling models. Kingma et al. (2014) introduced a stacked semi-supervised generative model, which combines a generative classifier with the latent representation generated by the encoder. Generative Adversarial Networks (GANs) (Goodfellow et al., 2020) have also been explored for semi-supervised learning (Odena, 2016). Besides generative models, graph-based methods have been developed to model data relationships, aiding semi-supervised

learning (Luo et al., 2018). The pseudo-labelling method (Lee, 2013), which trains a model on the labelled data and then uses the model to predict the labels of unlabelled data, has been widely verified in semi-supervised learning for many downstream tasks (Chen et al., 2024; Liu et al., 2022; Chen et al., 2023b). The basic idea is to use the model's predictions as pseudo-labels for unlabelled data to train the model with labelled data. MeanTeacher (Tarvainen & Valpola, 2017) introduced a consistency loss to enforce the model to be stable under small perturbations of the input data. FixMatch (Sohn et al., 2020) used strong data augmentations as the perturbation and introduced a threshold to filter out low-confidence pseudo labels. Based on the FixMatch framework, the following works such as FlexMatch (Zhang et al., 2021) and SoftMatch (Chen et al., 2023a) focused on improving the filtering mechanism of pseudo labels. Li et al. (2024) proposed a reward estimation algorithm to improve the quality of pseudo labels.

**Normalisation** techniques improve the training stability and convergence of deep learning models. BN (Ioffe & Szegedy, 2015) dominates the choice of normalisation techniques in convolutional neural networks. To solve the biased statistics estimation issue of BN with small batch sizes, Wu & He (2018) proposed GN which divides the channels into groups to calculate the mean and variance without the dependency on batch size. Transformers (Vaswani et al., 2017) demonstrated significant enhancements to neural language processing and computer vision. Transformers adopt LN which calculates the mean and variance of each data sample. In semi-supervised learning, most methods adopt the normalisation layer which is used in the corresponding fully-supervised models. Zajac et al. (2019) proposes to split the statistics calculation for data in different domains. EMANorm (Cai et al., 2021) replaced the BN in the teacher model of a teacher-student framework with an exponential moving average normalisation layer by calculating the mean and variance based on the student's statistics.

In real-world semi-supervised learning scenarios, the distribution of the unlabelled data subset is often uncertain. Without labels, it is difficult to effectively separate data from different distributions. This paper finds that the commonly used BN carries a significant risk of performance degradation in such a case.

## 3 METHOD

In this section, we first introduce two groups of widely adopted normalisation operations — batch-dependent normalisation such as BN, and batch-independent normalisation such as GN and LN. We use normalisation layers in image processing as an example in this section. The description of our proposed enhancement follows.

### 3.1 PRELIMINARIES

Two data subsets $\mathcal{D}^l$, and $\mathcal{D}^u$ are given for model optimisation in semi-supervised learning, where $\mathcal{D}^l = \{\mathcal{X}^l, \mathcal{Y}^l\}$ is the subset with available ground truth label $\mathcal{Y}^l$. $\mathcal{D}^u = \{\mathcal{X}^u, \mathcal{Y}^u\}$ is the unlabelled subset, but $\mathcal{Y}^u$ is unavailable during training.

### 3.2 NORMALISATION FORMULATION

The initial operation of most normalisation layers is shifting and scaling the input tensor to make it have zero mean and unit standard deviation:

$$o = \frac{x - \mu}{\sigma}, \tag{1}$$

where $x \in \mathbb{R}^{B \times C \times H \times W}$ is the input tensor, $o \in \mathbb{R}^{B \times C \times H \times W}$ is the normalised tensor, $\mu$ and $\sigma$ are the two statistics, *i.e.*, the mean and standard deviation, calculated from $x$. With the learnable affine transformation parameters $\gamma$ and $\beta$, the output tensor $o$ can be further scaled and shifted:

$$o = \frac{x - \mu}{\sigma} * \gamma + \beta. \tag{2}$$

The statistics calculation formulas are:

$$\mu = \frac{\sum_{i \in \mathcal{S}} x_i}{||\mathcal{S}||_0}, \quad \sigma = \sqrt{\frac{\sum_{i \in \mathcal{S}} (x_i - \mu)^2}{||\mathcal{S}||_0}}, \tag{3}$$

where $\mathcal{S}$ is the set of indices of the elements for calculating the statistics, and $||\mathcal{S}||_0$ is the number of elements in $\mathcal{S}$. When normalising an element $x_k$ in the input tensor, the difference between various normalisation layer types lies in which elements of the input $x$ are involved when calculating the statistics $\mu_k$ and $\sigma_k$ for $x_k$. The batch-dependent normalisation means that the elements of different images in the minibatch are involved in the statistics calculation. For example, in BN, the statistics are calculated over the minibatch dimension ($B$), and additional shape dimensions such as height and width ($H \times W$). In this case, the shape of $\mu$ and $\sigma$ is the same as the channel number $C$ of the input tensor $x$. Thus, $\mu_k$ and $\sigma_k$ are calculated with all the elements in the same channel as the $x_k$.

In contrast, batch-independent normalisation means that only the elements of the same image are involved. For example, in GN, the feature channels are divided into several groups, and the statistics are calculated over the group dimension. In such a way, the statistics of each data sample in GN are independent, which is more suitable for training with a small batch size. LN is a special case of GN, where the number of groups is equal to the number of feature channels.

### 3.3 DOES BATCH-INDEPENDENT NORMALISATION OUTPERFORM BATCH-DEPENDENT NORMALISATION?

The answer is YES and NO.

First, we find that GN is more robust when the distributions of $\mathcal{Y}^l$ and $\mathcal{Y}^u$ are different. We conduct experiments in semi-supervised image classification with the state-of-the-art SoftMatch (Chen et al., 2023a). The backbone is two ResNet-50s (He et al., 2016) with BN and GN respectively. The training data is CIFAR-100 (Krizhevsky, 2009). We follow the setting in RobustSSL (Jia et al., 2024) to manually split the categories in CIFAR-100 into two groups — in-distribution and out-of-distribution. $\mathcal{D}^u$ is constructed by images from the in-distribution and out-of-distribution categories with different ratios $r$. The larger $r$ is, the more different the distributions of $\mathcal{Y}^l$ and $\mathcal{Y}^u$ are. As shown in Fig. 2, when $r$ increases, the performance of BN declines significantly. One of the reasons is that with a large $r$, there are less in-distribution samples that can be used for model training. However, comparing the performance of BN with GN indicates that less in-distribution data is not the only reason. The performance of BN decreases more sharply. Such a phenomenon is attributed to a biased estimation of $\mu$ and $\sigma$ with out-of-distribution data. The biased estimation leads to an unstable upper bound of the gradient's difference between the two steps, which makes the optimisation unstable. More details are provided in the supplementary material (Appendix A). The calculation of the statistics in GN is independent of the batch data, which naturally alleviates this issue.

Secondly, we find that BN can be regained straightforwardly by calculating $\mu$ and $\sigma$ separately within different distributions. Each minibatch is divided into several parts according to the ground truth category labels and the image augmentations. The statistics in Eq. (2) are calculated separately for each part. For example, we split the training data into three parts: weakly-augmented in-distribution data, strong-augmented in-distribution data, and out-of-distribution data. Notably, we only use the real ground truth $\mathcal{Y}^u$ for analysis, we do not use it in our proposed method which will be introduced later. The baseline model with the regained BN (SepBN) sees a significant improvement, as shown in Fig. 2. Notably, SepBN surpasses GN when $r$ is small. We believe that the reason is that the randomness to a certain extent in the batch-dependent statistics calculation is a good regularisation to the model training, especially for semi-supervised learning which requires a

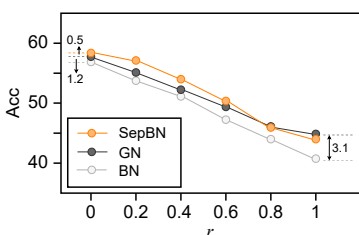

Figure 2: Performance on CIFAR100 of different normalisation with various $r$.

good generalisation ability to produce high-quality pseudo labels. For a certain image $x$, the other images in the minibatch at different iteration steps are different. Consequently, the statistics $\mu$ and $\sigma$ calculated from different minibatch are different in BN. However, GN calculates the statistics independently for each image, which is less random.

Thus, inspired by the above analysis, we propose a new normalisation layer called Shuffle Group/Layer Normalisation (SGN/SLN) to combine the advantages of BN and GN/LN without introducing additional parameters and computing overload.

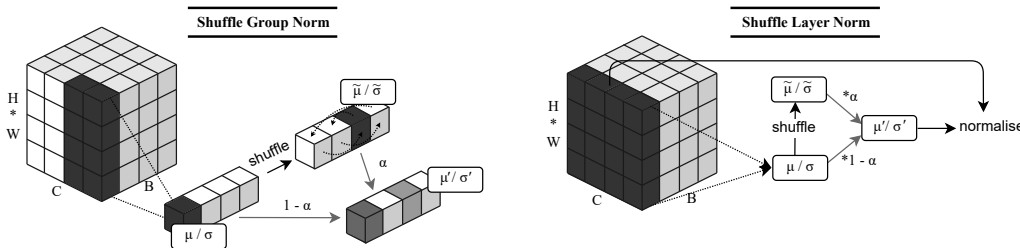

Figure 3: Demonstration of the proposed SGN/SLN.

### 3.4 SHUFFLE GROUP/LAYER NORMALISATION

We introduce more randomness into the statistics calculation of GN/LN to construct the SGN/SLN. The key operation is a shuffling after calculating the statistics:

$$\tilde{\mu} = \mu[I], \quad \tilde{\sigma} = \sigma[I], \quad I = \text{shuffle}(\{0, 1, \cdots, B-1\}), \quad (4)$$

where $I$ is the shuffled index, $B$ is the batch size, and $\tilde{\mu}$ and $\tilde{\sigma}$ are the shuffled statistics. The shuffling operation is performed on the batch dimension. The shuffled statistics are then used as a perturbation to the original statistics with a factor $\alpha$:

$$\mu' = (1 - \alpha)\mu + \alpha\tilde{\mu}, \quad \sigma' = (1 - \alpha)\sigma + \alpha\tilde{\sigma}. \quad (5)$$

The perturbed statistics $\mu'$ and $\sigma'$ are then used to normalise the input tensor $x$. Fig. 3 shows the workflow of the proposed SGN/SLN. The pseudo-code is described in Algorithm 1.

During the inference stage, the statistics are perturbed with the moving average $\overline{\mu}$ and $\overline{\sigma}$ of the statistics calculated during the training stage to stabilise the inference:

$$\mu' = (1 - \alpha)\mu + \alpha\overline{\mu}, \quad \sigma' = (1 - \alpha)\sigma + \alpha\overline{\sigma}. \quad (6)$$

By performing the shuffling operation in the calculation of the statistics, we introduce more randomness into the normalisation layer. It can be regarded as a controllable regularisation to improve the generalisation ability of the model. Consequently, the confirmation bias issue in semi-supervised learning is alleviated. Notably, this is not a simple linear combination of BN and GN/LN. Firstly, the randomness for different samples in a minibatch varies, as $\tilde{\mu}$ and $\tilde{\sigma}$ are distinct for each sample. The shuffled index $I$ also differs across layers. Most importantly, the GN/LN in pretrained foundation models can be directly replaced by SGN/SLN and initialised using the pretrained parameters, as no additional learnable parameters are introduced in our method. There is no pretrained model containing both BN and LN/GN that can be used for such a linear combination.

## 4 EXPERIMENTS

In this section, experiments are conducted to evaluate the proposed SGN/SLN in semi-supervised learning. The proposed method is implemented with the PyTorch framework (Paszke et al., 2019). The code can be found in the public repository after publishing.

### 4.1 ROBUST SEMI-SUPERVISED LEARNING SETTING

We first evaluate different normalisation layers in semi-supervised learning with the inconsistency label distributions setting to demonstrate that the proposed *SLN/SGN can make semi-supervised algorithms more robust*.

#### 4.1.1 DATASETS AND IMPLEMENTATION DETAILS

The dataset and baseline source code used in this subsection are published by Jia et al. (2024). We conduct experiments on CIFAR10 (Krizhevsky, 2009), CIFAR100 (Krizhevsky, 2009), and Semi-Supervised INaturalist-Aves (SemiAves) (Su & Maji, 2021). Here, we use CIFAR100 as an example

**Algorithm 1** The operation of the SGN.

```
"""
x: Tensor(B*C*H*W), input tensor
group_num: int, the number of groups, if group_num equals to the channel numbers, it is
    equivalent to LN
alpha: float, the factor of perturbation,
gamma: Tensor(Optional), the scaling factor
beta: Tensor(Optional), the shifting factor
m: float, the moving average momentum
"""
class ShuffleGN(nn.Module):
    def __init__(self, group_num, alpha, gamma=None, beta=None, m=0.1):
        super(ShuffleGN, self).__init__()
        self.alpha = alpha
        self.gamma = gamma
        self.beta = beta
        self.m = m
        self.eps = 1e-5
        self.register_buffer('running_mu', torch.zeros(group_num))
        self.register_buffer('running_var', torch.zeros(group_num))

    def forward(self, x):
        groups = torch.chunk(x, group_num, dim=1)
        grouped_x = torch.stack(groups, dim=1) # B * group_num * C/group_num * H * W
        mu = torch.mean(grouped_x, dim=[2, 3, 4], keepdim=True) # B * group_num * 1 * 1 * 1
        var = torch.var(grouped_x, dim=[2, 3, 4], keepdim=True, unbiased=False) # B *
            group_num * 1 * 1 * 1
        # update the running statistics
        self.running_mu = (1 - self.m) * self.running_mu + self.m * mu.mean(dim=0).squeeze()
        self.running_var = (1 - self.m) * self.running_var + self.m * var.mean(dim=0).squeeze
            ()
        if self.training:
            # shuffle the batch dimension
            batch_size = x.size(0)
            shuffle_index = torch.randperm(batch_size)
            shuffle_mu = mu[shuffle_index]
            shuffle_var = var[shuffle_index]
            # perturb the statistics
            perturbed_mu = (1 - alpha) * mu + alpha * shuffle_mu
            perturbed_var = (1 - alpha) * var + alpha * shuffle_var
        else:
            perturbed_mu = (1 - alpha) * mu + alpha * self.running_mu
            perturbed_var = (1 - alpha) * var + alpha * self.running_var

        # normalise the input tensor
        x = (x - perturbed_mu) / torch.sqrt(perturbed_var + eps)
        # scale and shift if needed
        if gamma is not None and beta is not None:
            x = x * gamma + beta
        return x
```

to explain this setting. CIFAR100 contains 60,000 $32 \times 32$ colour images in 100 classes, with 50000 training images and 10000 test images. The predefined categories are divided into two groups — 50 categories for in-distribution data, and the other 50 categories for out-of-distribution data. The unlabelled set is mixed with in-distribution and out-of-distribution data with different ratios $r$. $r = 0$ means that there is no out-of-distribution data. The model is trained to conduct 50 in-distribution data classifications. The SOTA semi-supervised learning algorithm SoftMatch (Chen et al., 2023a) serves as the baseline in this setting.

### 4.1.2 PERFORMANCE

We evaluate BN, GN, and SGN. The results are shown in Tabs. 1 to 3. The performance of the naive BN drops significantly when the ratio of out-of-distribution data increases. For example in Tab. 1, as the ratio $r$ increases from 0.0 to 1.0, the accuracy of BN sees a drop of 14.46%. GN performs well in this scenario. The accuracy of GN drops by only 12.55%. Our SGN outperforms all the other normalisation layers in the inconsistency label distribution setting as it resolves the biased statistic estimation problem in BN (only drops 11.46%), and introduces the randomness regularisation from GN/LN.

### 4.2 SEMI-SUPERVISED LEARNING SETTING

In addition to the robust semi-supervised learning setting, we conduct extensive traditional semi-supervised learning experiments on datasets from three modalities including image, text, and audio.

Table 1: The performance on CIFAR-100 with different robust ratios $r$.

| $r$ | 0.0 | 0.2 | 0.4 | 0.6 | 0.8 | 1.0 |
|-----|-----|-----|-----|-----|-----|-----|
| BN | 56.11 | 53.98 | 51.49 | 48.43 | 44.33 | 41.65 |
| GN | 57.33 | 55.13 | 52.40 | 49.73 | 46.29 | 44.78 |
| SGN | **59.52** | **57.64** | **54.56** | **51.56** | **49.40** | **48.06** |

Table 2: The performance on CIFAR-10 with different robust ratios $r$.

| $r$ | 0.0 | 0.2 | 0.4 | 0.6 | 0.8 | 1.0 |
|-----|-----|-----|-----|-----|-----|-----|
| BN | 89.86 | 88.09 | 86.45 | 84.86 | 81.80 | 78.23 |
| GN | 89.11 | 88.06 | 86.49 | 84.88 | 82.72 | 81.33 |
| SGN | **90.43** | **89.27** | **87.71** | **86.02** | **83.78** | **82.49** |

Table 3: The performance on SemiAves with different robust ratio $r$.

| $r$ | 0.0 | 0.2 | 0.4 | 0.6 | 0.8 | 1.0 |
|-----|-----|-----|-----|-----|-----|-----|
| BN | 28.82 | 28.08 | 25.72 | 24.21 | 22.17 | 20.65 |
| GN | 34.66 | 32.02 | 29.80 | 28.47 | 26.63 | 26.40 |
| SGN | **37.69** | **35.56** | **33.09** | **31.13** | **28.98** | **27.62** |

Table 4: The performance on semantic segmentation (the metric is mIoU).

| Norm. Layer | GN | SGN |
|-------------|-----|-----|
| Cityscapes (1/30) | 67.00 | **68.10** |
| PascalVOC (1/16) | 74.91 | **75.54** |

The experiments in this subsection show that *the proposed normalisation layer is a better option for normalisation layers in semi-supervised learning*.

### 4.2.1 DATASETS AND IMPLEMENTATION DETAILS

The datasets used in this setting include:

**Image Datasets:** CIFAR100, as described in Sec. 4.1.1. SemiAves (Su & Maji, 2021) contains 1000 species of birds sampled from the iNat-2018 dataset (Horn et al., 2018) for a total of nearly 150k images. The STL10 dataset is for the unsupervised learning research. In particular, fewer labelled training examples and a very large set of unlabeled examples are available for training.

**Text Datasets:** Amazon Review dataset (Majumder et al., 2020) is a sentiment classification dataset which contains 600k reviews for training and 130k reviews for testing. Yahoo Answer dataset (Zhang et al., 2015) contains 140k training samples and 6k testing samples from 10 classes, which is for the topic classification. The Yelp Review dataset is a sentiment classification dataset which contains 650k training samples and 500k testing samples. In this paper, we use the subsets drawn by the USB framework for the experiments.

**Audio Datasets:** ESC-50 (Piczak, 2015) is a dataset for environmental sound classification, which contains 2000 samples from 50 classes. UrbanSound8K dataset (Salamon et al., 2014) contains 8732 labelled sound clips (¡=4s) from ten classes. FSDNoisy dataset (Fonseca et al., 2019) is a dataset for sound event classification, which contains 17k samples from 20 classes. GTZAN dataset (Tzanetakis, 2001) is a dataset for music genre classification. We use the dataset resampled by the USB framework which contains 7k samples for training, 1.5k for validation/testing in our experiment.

SoftMatch still serves as the baseline in this setting. We use the proposed SLN/SGN to replace the original normalisation layers in the backbone. All results are averaged accuracy produced with 3 different random seeds (0/1/2).

### 4.2.2 PERFORMANCE

**Semi-supervised Image Classification:** We use the abovementioned three image datasets to evaluate SLN in the image modality. The backbone of the baseline model is the Vision Transformer (Dosovitskiy et al., 2021) with LN. The results of the semi-supervised image classification experiments are shown in Tab. 5. The proposed SLN boosts the performance of the baseline Soft-Match on all three datasets significantly. On the STL10 dataset, our method boosts the accuracy by **3.72%** compared to the baseline.

**Semi-supervised Text Classification:** We replace the layer normalisation in the BeRT (Devlin et al., 2019) backbone with the proposed SLN and conduct experiments on three text datasets. The results are shown in Tab. 6. The proposed SLN consistently outperforms the baseline SoftMatch on

Table 5: The performance on the image modality. The number in the bracket is the number of labelled data.

| Methods | CIFAR100 (200) | SemiAves (1000) | STL10 (40) |
|---|---|---|---|
| Labelled-Only | 64.12 | - | 81.00 |
| Pseudo Label (Lee, 2013) | 66.01 | 35.40 | 80.86 |
| MeanTeacher (Tarvainen & Valpola, 2017) | 64.53 | 39.30 | 81.33 |
| MixMatch (Berthelot et al., 2019) | 61.78 | 34.73 | 41.23 |
| FixMatch (Sohn et al., 2020) | 70.40 | 46.20 | 83.85 |
| FlexMatch (Zhang et al., 2021) | 73.24 | - | 85.60 |
| CoMatch (Li et al., 2021) | 64.92 | - | 84.88 |
| SoftMatch (Chen et al., 2023a) | 77.55 | 46.05 | 87.67 |
| SoftMatch (w/ ours) | **78.55** | **47.10** | **91.39** |

Table 6: The performance on the text modality. The number in the bracket is the number of labelled data.

| Methods | Amazon Review (250) | Yahoo Answers (500) | Yelp Review (250) |
|---|---|---|---|
| Labelled-Only | 47.69 | 62.57 | 48.78 |
| Pseudo Label (Lee, 2013) | 46.55 | 62.30 | 45.49 |
| MeanTeacher (Tarvainen & Valpola, 2017) | 47.86 | 62.91 | 49.40 |
| MixMatch (Berthelot et al., 2019) | 40.46 | 64.25 | 46.02 |
| FixMatch (Sohn et al., 2020) | 52.39 | 66.97 | 53.48 |
| FlexMatch (Zhang et al., 2021) | 54.27 | 64.39 | 56.65 |
| CoMatch (Li et al., 2021) | 51.24 | 66.52 | 54.60 |
| SoftMatch (Chen et al., 2023a) | 55.23 | 67.30 | 56.65 |
| SoftMatch (w/ ours) | **55.90** | **68.51** | **57.31** |

Table 7: The performance on the audio modality. The number in the bracket is the number of labelled data.

| Methods | ESC50 (250) | GTZAN (100) | FSDNoisy (1773) | Urbansound8K (400) |
|---|---|---|---|---|
| Labelled-Only | 50.17 | 47.27 | 65.26 | 72.40 |
| Pseudo Label (Lee, 2013) | 49.92 | 46.53 | 62.16 | 70.17 |
| MeanTeacher (Tarvainen & Valpola, 2017) | 48.17 | 49.84 | 66.56 | 70.97 |
| MixMatch (Berthelot et al., 2019) | 40.00 | 25.36 | 46.85 | 58.62 |
| FixMatch (Sohn et al., 2020) | 56.40 | 58.50 | 69.00 | 79.17 |
| FlexMatch (Zhang et al., 2021) | 60.67 | 49.29 | 72.65 | 76.30 |
| CoMatch (Li et al., 2021) | 59.33 | 59.51 | 71.88 | 79.81 |
| SoftMatch (Chen et al., 2023a) | 67.00 | 68.71 | 72.22 | 77.18 |
| SoftMatch (w/ ours) | **67.42** | **69.73** | **72.93** | **80.25** |

all three datasets. For example, on the Yahoo Answers dataset, we achieve a **1.21%** improvement in accuracy compared to the baseline.

**Semi-supervised Audio Classification:** On the audio modality, the baseline model with the backbone HuBert (Hsu et al., 2021) armed with the proposed SLN achieves state-of-the-art performance on all four datasets as shown in Tab. 7. The proposed SLN also outperforms the baseline SoftMatch on all datasets. Notably, on the Urbansound8K dataset, the model with the SLN achieves a **3.07%** improvement in accuracy compared to the baseline.

### 4.3 MORE TASKS

To demonstrate the generalisation of SLN/SGN, we conduct experiments in semi-supervised semantic segmentation first. The state-of-the-art semi-supervised semantic segmentation algorithm, PrevMatch (Shin et al., 2024), serves as the baseline model. All normalisation layers in the backbone ResNet-50 (GN) are replaced by the proposed SGN, and the results are reported in Tab. 4. The proposed SGN consistently improves performance. As the normalisation layers in object de-

Table 8: The performance on ImageNet of fully supervised image classification.

| Norm. Layer | GN/LN | SGN/SLN |
|---|---|---|
| ResNet 50 | 76.0 (GN) | **76.3** (SGN) |
| Swin-T | 80.8 (LN) | **81.1** (SLN) |

Table 9: Ablation study of the $\alpha$.

| $\alpha$ | 0.0 (baseline) | 0.1 | 0.3 | 0.4 | 0.5 | 0.7 | 0.9 |
|---|---|---|---|---|---|---|---|
| Acc. | 77.55 | 78.23 | 79.02 | 78.55 | 77.33 | 74.83 | 66.31 |

tection models (Liu et al., 2021) are usually frozen, it is not compatible to evaluate the proposed normalisation layer in it.

In addition, we evaluate the proposed SGN/SLN in fully-supervised image classification on Ima-geNet dataset (Deng et al., 2009). As shown in Tab. 8, both convolutional neural networks and vision transformers benefit from SGN/SLN.

## 5 ABLATION STUDY

Comparing the performance of the model w/ and w/o the proposed SLN/SGN in Tabs. 5 to 7 suggests that SLN/SGN is effective in improving the performance of semi-supervised learning models.

In addition, we use the CIFAR100 (200) as an example to ablate the $\alpha$ used in the shuffle normal-isation layer and report the results in Tab. 9. The baseline model without our method is equivalent to $\alpha = 0$. We observe a peek in performance at $\alpha = 0.3$. With the increase of $\alpha$, the performance of the model decreases. This suggests that the randomness introduced by the shuffle normalisation layer is important for the model, but too much randomness can hurt the performance of the model.

We also discuss using the proposed shuffle operation only during inference in Appendix C, and the results indicate that it is not effective.

## 6 ANALYSIS

In this section, we first discuss the randomness in the normalisation layer. Then we analyse the model w/ and w/o the proposed SLN as an example to reveal the reasons why the proposed normal-isation benefits models.

### 6.1 RANDOMNESS IN NORMALISATION

In addition to Fig. 2, we show the performance of SepBN in CIFAR10 on Fig. 4. SepBN performs better than GN when the ratio $r$ is small, and the accuracy drop is smaller than BN. Comparing the results of GN and SepBN with BN reveals that the biased estimation of the statistics in BN is harmful to semi-supervised learning. Comparing the results of SepBN with GN indicates that the randomness regularisation in BN is helpful to semi-supervised learning.

### 6.2 HESSIAN EIGENVALUE ANALYSIS

The definition of the Hessian matrix is a square matrix of second-order partial derivatives of a scalar-valued function. In this paper, we use the Hessian matrix to analyze the curvature of the loss func-tion. We calculate the Hessian matrix of the loss function with respect to the model's input. By analysing such a Hessian, we can analyse whether the loss function landscape is sharp or smooth around the data point in the input space. As the calculation of the Hessian is computationally expen-sive, we use the Lanczosn algorithm (Lanczos, 1950) to estimate the top eigenvalues of the Hessian matrix. The top eigenvalues of the Hessian matrix can be used to estimate the curvature of the loss function:

$$\lambda_{\max} = \max \boldsymbol{\lambda}(\boldsymbol{H}_{\mathcal{L}}), \tag{7}$$

where $\boldsymbol{H}_{\mathcal{L}}$ is the Hessian matrix of the loss function $\mathcal{L}$. $\boldsymbol{\lambda}$ calculates the set of eigenvalues of the Hessian matrix. We compare the $\lambda_{\max}^{\text{SLN}}$ and $\lambda_{\max}^{\text{w/o SLN}}$ and report $\Delta\lambda_{\max} = \lambda_{\max}^{\text{SLN}} - \lambda_{\max}^{\text{w/o SLN}}$, which is averaged over all the data points in the test set, in Tab. 10a. The results show that all $\Delta\lambda_{\max}$ are negative, which suggests that the loss function landscape is smoother when SLN is used.

Table 10: a) Analysis with the maximum eigenvalue difference $\Delta\lambda_{max}$ for models w/ and w/o our method. The model's parameter is the best checkpoint on the test set. b) Analysis of $\Delta\lambda_{max}$ at different training epochs.

(a)

|  | CIFAR100 | SemiAves | STL10 |
|---|---|---|---|
| $\Delta\lambda_{max}$ | -132.5 | -14.9 | -3.7 |

(b)

| Epochs | 20 | 40 | 60 | 80 | 100 |
|---|---|---|---|---|---|
| $\Delta\lambda_{max}$ | -258.5 | -168.3 | -148.7 | -158.3 | -160.5 |

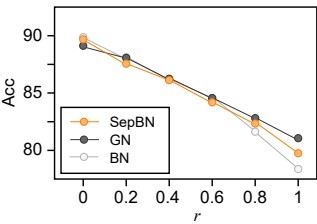

Figure 4: Performance on CI-FAR10 of different normalisation with various $r$.

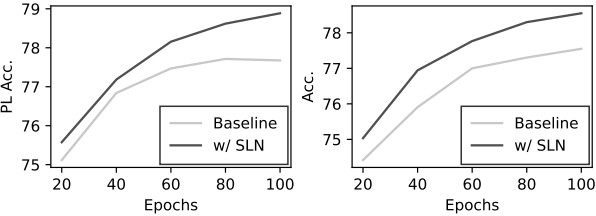

Figure 5: left) The accuracy of the baseline model and the model with our method on the validation set. right) The accuracy of the pseudo-labels during training.

Consequently, the model with a smoother loss function landscape is more robust to different data points in the input space, leading to better pseudo-label quality and superior performance. We also show the $\Delta\lambda_{max}$ of a model at different training epochs in Tab. 10b. The model with SLN has a consistently smaller $\Delta\lambda_{max}$ than the model without SLN.

### 6.3 PSEUDO-LABEL QUALITY ANALYSIS

A better pseudo-label quality can lead to a better model performance. We plot the accuracy of the pseudo labels generated by the model w/ and w/o the SLN in Fig. 5. The results show that the model with the SLN has a higher pseudo-label accuracy than the model without the SLN. This suggests that the SLN can improve the pseudo-label quality, which leads to better model performance.

### 6.4 COMPARING PERFORMANCE IN FULLY SUPERVISED LEARNING AND SEMI-SUPERVISED LEARNING

We report the performance of SLN in fully supervised learning in Tab. 8. Compared with semi-supervised learning, the performance gain in fully supervised learning is relatively limited. The reason is that the large number of labels in fully supervised learning provides strong supervision and vivid data points, reducing the reliance on stochastic regularisation in the model. In contrast, semi-supervised learning only uses very few labelled data. In the early stages of training, the model quickly fits the small amount of labelled data, leading to a sharp loss landscape. The absence of randomness regularisation in LN/GN exacerbates this problem. SLN/SGN introduce some controllable randomness into the model's optimisation which benefits the optimisation of the model.

### 7 CONCLUSION

In this paper, we studied the normalisation layers in semi-supervised learning. We found that the widely used normalisation layers, such as BN, GN, and LN are suboptimal in semi-supervised learning. By modifying GN and LN to introduce additional randomness, SLN/SGN were proposed to improve models' robustness and performance without adding extra parameters. Extensive experiments on various modalities suggest it is a better option for normalisation layers in semi-supervised learning tasks. Inspired by our findings on the importance of stochastic regularisation in the normalisation layers to semi-supervised learning, in future work, we could analyse more modules within neural networks to explore whether the discoveries in this paper can further improve the performance of semi-supervised learning algorithms.

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

## A    DERIVATION: BIASED ESTIMATES LEAD TO AN UNSTABLE OPTIMISATION

In this section, we derive why biased estimates lead to an unstable optimisation.

Suppose we have a one-layer neural network consisting of a linear layer with a weight $w$ and a bias $b$:

$$o = wx + b, \tag{8}$$

where $x$ is the input and $o$ is the output. To optimise this neural network, a loss function $\mathcal{L}$ with a non-linear activation function $f$ should be minimised:

$$\mathrm{argmin}_{w,b} \mathcal{L}(f(wx + b)). \tag{9}$$

Considering a dataset $p(x)$, we randomly draw samples from it to optimise the neural network with Eq. (9). The gradient of $\mathcal{L}$ w.r.t. the weight $w$ is:

$$\mathbb{E}_{x \sim p(x)}[\nabla_w \mathcal{L}] = \mathbb{E}_{x \sim p(x)}[\frac{\partial \mathcal{L}}{\partial f} \frac{\partial f}{\partial o} x]. \tag{10}$$

Once the weight $w$ is optimised for one step $\epsilon$, the gradient is:

$$\mathbb{E}_{x \sim p(x)}[\nabla_{w+\epsilon} \mathcal{L}] = \mathbb{E}_{x \sim p(x)}[\frac{\partial \mathcal{L}}{\partial f} \frac{\partial f}{\partial o'} x], \tag{11}$$

where $o' = (w + \epsilon)x + b$. The difference between the two gradients is:

$$\mathbb{E}_{x \sim p(x)}[\nabla_w \mathcal{L} - \nabla_{w+\epsilon} \mathcal{L}] = \mathbb{E}_{x \sim p(x)}[(\frac{\partial \mathcal{L}}{\partial f} \frac{\partial f}{\partial o} - \frac{\partial \mathcal{L}}{\partial f} \frac{\partial f}{\partial o'})x]. \tag{12}$$

Usually, $\frac{\partial \mathcal{L}}{\partial f} \frac{\partial f}{\partial o}$ is bounded. For example, in binary classification tasks, the cross entropy serves as $\mathcal{L}$ and $f$ is the sigmoid function. In this case, the range of $\frac{\partial \mathcal{L}}{\partial f} \frac{\partial f}{\partial o}$ is $[-1, 1]$. Consequently, $\delta = \frac{\partial \mathcal{L}}{\partial f} \frac{\partial f}{\partial o} - \frac{\partial \mathcal{L}}{\partial f} \frac{\partial f}{\partial o'}$ should be stable during training. For some loss functions with unbounded gradients, good model initialisation techniques can usually ensure stability for this term (Su, 2019). As a result, the stability of the gradient is highly dependent on the input $x$.

With the Cauchy–Schwarz inequality and Eq. (12), the upper bound of Eq. (12) is:

$$||\mathbb{E}_{x \sim p(x)}[\delta x]||_2 \leq \sqrt{\mathbb{E}_{x \sim p(x)}[\delta^2]} \times \sqrt{\mathbb{E}_{x \sim p(x)}[x \otimes x]}. \tag{13}$$

To get a stable gradient, *i.e.*, a smaller $||\mathbb{E}_{x \sim p(x)}[\delta x]||_2$, normalise $x$ to minimise the upper bound is a feasible way.

With BN, the input $x$ is shifted by the mean $\mu = \mathbb{E}_{x \sim p(x)}[x]$ and scaled by the standard deviation $\sigma = \sqrt{\mathbb{E}_{x \sim p(x)}[(x - \mu) \otimes (x - \mu)]}$:

$$\hat{x} = \frac{x - \mu}{\sigma}. \tag{14}$$

Thus,

$$\mathbb{E}_{x \sim p(x)}[\hat{x} \otimes \hat{x}] = \frac{\mathbb{E}_{x \sim p(x)}[(x - \mu) \otimes (x - \mu)]}{\sigma \otimes \sigma} = \mathbf{1}. \tag{15}$$

As a result, BN normalise the input $x$ to eliminate $\sqrt{\mathbb{E}_{x \sim p(x)}[x \otimes x]}$ in Eq. (13), thereby assuring a small upper bound of the gradient difference to stabilise the optimisation. Although the $\mu$ and $\sigma$ are estimated within each minibatch in practice, a small bias won't significantly change the upper bound. However, If there are too many out-of-distribution data $x' \sim q(x)$ in a minibatch, the estimated $\mu'$ and $\sigma'$ are significantly biased, therefore yielding an unstable $\sqrt{\mathbb{E}_{x \sim p(x)}[x \otimes x]}$ in Eq. (13). It inevitably hurts the optimisation of the neural network. For LN/GN, the upper bound is stable as the estimated $\mu'$ and $\sigma'$ are independent of the other samples in the same minibatch.

## B    Implementation Details

For each backbone model in the main text, we replace all the normalisation layers with the corresponding normalisation layers we proposed. If the original normalisation layer is LN, we replace it with SLN; if it is GN, we replace it with SGN. As for CNNs, many baseline models use BN, so we replace the backbone with one pre-trained[1] using GN as the baseline model to compare with our method.

The main hyperparameter of our method is $\alpha$. For the experiments in the robust semi-supervised learning setting (Sec. 4.1), $\alpha = 0.4$. For the experiments in the semi-supervised setting(Sec. 4.2), we use $0.4$ for CIFAR-100, ESC50, GTZAN, FSDNoisy and Urbansound8K; $0.1$ for SemiAves, STL10, Amazon Review, Yahoo Answers, and Yelp Review.

## C    Can We Shuffle the Statistic Values During Only Inference Rather than Training?

The conclusion is negative. The performance of the model can only be improved by incorporating the randomness regularisation proposed in this paper during training. Adding it only in the inference phase does not allow an untrained model to adapt well to such random perturbations, which may result in a performance drop. For example, when we evaluated the model (LN) trained on the CIFAR100 (200) dataset and introduced perturbations during testing, the model's performance dropped from 77.55 to 77.36.

---

[1]https://github.com/ppwwyyxx/GroupNorm-reproduce/releases

