# OpenReview forum: "ShuffleNorm: A Better Normalization for Semi-supervised Learning"
_ICLR.cc/2025/Conference — ICLR 2025 Conference Withdrawn Submission_

### Official Review · Reviewer_wLX2 · 2024-10-30

**Soundness:** 3
**Presentation:** 1
**Contribution:** 2
**Rating:** 5
**Confidence:** 3

**Summary:**

The paper proposes a novel normalization technique called Shuffle Layer Normalization (SLN) to address the challenges of applying normalization layers in semi-supervised learning.  The authors identify that Batch Normalization (BN) can cause performance degradation when labeled and unlabeled data have mismatched label distributions due to biased statistical estimation, while Group/Layer Normalization (GN/LN) lacks the stochastic regularization provided by BN, leading to weaker generalization.

**Strengths:**

The authors effectively identify the limitations of existing normalization techniques in semi-supervised learning, particularly the performance degradation of BN with mismatched label distributions and the lack of stochastic regularization in GN/LN. The proposed SLN/SGN method offers a novel approach to address these limitations by introducing controllable randomness into the normalization process.

**Weaknesses:**

1) The core idea of introducing randomness into normalization layers is not entirely new. Other techniques, such as dropout, have explored similar concepts. The dropout introduce randomness to enhance the generalization ability of the model. Randomly drops out neurons (along with their connections) during training, forcing the network to learn more robust features that are not dependent on the presence of specific neurons. While this paper randomizes the normalization statistics by shuffling the order of samples in a batch. This introduces noise into the normalization process, preventing the model from relying too heavily on specific batch statistics.
2) The paper primarily relies on empirical evidence to support the effectiveness of SLN/SGN. There need a more rigorous theoretical analysis of why the proposed method works. How does the introduced randomness affect the optimization landscape? Does it lead to a smoother loss surface with fewer local minima, potentially explaining the improved performance?
3) The paper mainly compares SLN/SGN with BN and GN/LN. A more comprehensive comparison with other semi-supervised learning techniques, such as consistency regularization or pseudo-labeling methods, would provide a better understanding of the method's performance relative to the state-of-the-art. I recommed add comparision methods such as MixMatch, FixMatch and pseudo-labeling method.
4) While compatibility with pre-trained models is a strength, it also implies that the method's effectiveness may be limited when pre-trained models are not available or suitable. Based on the evaluation, the authors should discuss the implications of their findings for situations where appropriate pre-trained models are not readily available.

**Questions:**

1) The paper mentions a shuffling operation after calculating the statistics. Could the authors provide more details about this shuffling operation? Is it a random shuffle of the entire batch or are there specific patterns or constraints?
2) The paper introduces a perturbation factor 'a' to control the randomness. How was this factor determined for different datasets and modalities? Is there a theoretical or empirical justification for the optimal range of the perturbation factor?

---

### Official Review · Reviewer_LVtb · 2024-10-30

**Soundness:** 1
**Presentation:** 3
**Contribution:** 1
**Rating:** 1
**Confidence:** 3

**Summary:**

This paper examines the effectiveness of normalization layers in semi-supervised learning. It finds that Batch Normalization (BN) suffers performance degradation when labeled and unlabeled data have different label distributions. While Group/Layer Normalization (GN/LN) avoids this issue, it lacks the stochastic regularization of BN, hindering generalization and pseudo-label quality. The authors propose Shuffle Group/Layer Normalization (SGN/SLN), which introduces randomness into GN/LN without new parameters. Experiments across image, text, and audio datasets show SGN/SLN improves semi-supervised learning performance compared to BN and GN/LN, as well as enhancing the performance obtained by some models.  The authors analyze this improvement through the lens of randomness regularization and Hessian eigenvalue analysis which indicates SGN/SLN smooths the loss landscape.

**Strengths:**

* Clear identification and analysis of a problem: The paper clearly identifies the limitations of existing normalization techniques (BN and GN/LN) in the context of semi-supervised learning with mismatched label distributions between labeled and unlabeled data.  The analysis, supported by empirical evidence and theoretical justification, convincingly establishes the problem.

* Simple, yet effective solution: The proposed SGN/SLN method offers a straightforward modification to existing GN/LN layers. By introducing controlled randomness through shuffling, it addresses the identified limitations without adding complexity or parameters. This simplicity makes it easily adaptable to various models and frameworks.

**Weaknesses:**

1. All the tasks that are being assessed are classification tasks which diminishes the credibility of the actual results of the proposed method. The authors should also include estimation tasks (e.g. audio source separation) and generative tasks (e.g. machine translation) to get a better understanding whether the proposed layers work in a variety of tasks.

2. The authors include very basic baseline architectures in their experiments which also diminishes the value of their laims about improving the performance of several models. All the models used are some from 2021 papers and if anyone visits the latest state-of-the-art models for all theses tasks on any datasets are very different models that are being used. It could be the case that this regularization works for some basic transformer models under some training conditions on some specific tasks that help the models converge faster to some good solution but the effect will go away as more training steps are being completed. For instance, I can easily find over the internet multiple architectures the have way higher performance for semi-supervised training (e.g. see [A] for ~80% accuracy in semi-supervised training). In order for the authors to show significance in their results, then they need to get a SOTA architecture + loss + way of training and after they match it simply replace the normalization layers tos how that their method works.

3. Following up on my previous claim, it seems that when the authors tried to do that (see Table 8 with ResNet50 experiments on fully supervised Imagenet) then the results were not great at all which makes me even more skeptical about the validity of this method. Specifically, we already know that one can get over 80% top-1 accuracy on Imagenet with ResNet50 using various tricks [B] but the authors report that their regularization trick only achieves a minor increase 76 -> 76.3%.

[A] Xu, J.W. and Yeh, Y.R., 2024, July. Audio Classification with Semi-Supervised Contrastive Loss and Consistency Regularization. In 2024 IEEE 48th Annual Computers, Software, and Applications Conference (COMPSAC) (pp. 1770-1775). IEEE.

[B] Wightman, R., Touvron, H. and Jégou, H., 2021. Resnet strikes back: An improved training procedure in timm. arXiv preprint arXiv:2110.00476.

**Questions:**

Why do the authors intuitivvely think that their method works better?

---

### Official Review · Reviewer_YAL2 · 2024-11-04

**Soundness:** 3
**Presentation:** 3
**Contribution:** 3
**Rating:** 6
**Confidence:** 4

**Summary:**

The work identifies an issue with the batch-norm layer in the case of semi-supervised learning.
The authors postulate the loss in performance is due to mixing statistics of supervised and weakly-supervised samples, making the BN statistics inaccurate. The authors further postulate that while the layer/group normalization approaches don't suffer the same limitation, they lack some of the regularization achieved with randomness of batch statistics.
The author propose a method that corrects the standard batch-independent mu/sigma from layer/group normalization algorithms with a batch-dependent component that results from the shuffling of the samples inside the batch.

**Strengths:**

The strength of this paper is to identify a clear issue and propose a clear simple solution to a problem that can have an important potential impact. This is difficult combo-move to pull-off.
The experimental results are very convincing, almost too good to be true. I don't deny that they are true, just that the improvement shown by such a simple method is somewhat unexpected by my mental model at least. The improvement on STL10 and UrbanSound is massive.
The fact the paper presents experimental results across a large set of benchmarks, datasets and modality is also a big plus IMO.
Finally the fact the method is so simple, it doesn't require more parameters, and I understand can be added on top of pre-trained models (that were training using standard batch-norm / layer-norm) is also a huge plus.

**Weaknesses:**

The main claim that is motivated only intuitively, that local (layer/group ~ non batch statistics based) normalization methods are not proving the regularization that batch statistics provide, while credible, it's not very well substantiated.

Perhaps the authors could elaborate mode on this. Or possibly, do a deep literature search and cite a paper that presents such claim with clear substantiating material.

This is important IMO because it's the main reason to embark on this mission to bring the best of both words (all words) in the area of normalization layers

**Questions:**

Table 10 - I wonder if it makes sense to report the actual value of (of the EV) --or perhaps present the relative/ratio DeltaEV/EV0
without it I am not sure how much we can learn from that table

Also if possible address what I descrive under weakness

This is a good paper IMO.
If fully addressing my questions and concerns, I would be more than ready to up the rating!

---

### Note · Authors · 2025-01-11

I have read and agree with the venue's withdrawal policy on behalf of myself and my co-authors.